# The Neglected Liaison: Targeting Cancer Cell Metabolic Reprogramming Modifies the Composition of Non-Malignant Populations of the Tumor Microenvironment

**DOI:** 10.3390/cancers13215447

**Published:** 2021-10-29

**Authors:** Maria Iorio, Nikkitha Umesh Ganesh, Monica De Luise, Anna Maria Porcelli, Giuseppe Gasparre, Ivana Kurelac

**Affiliations:** 1Department of Medical and Surgical Sciences (DIMEC), University of Bologna, 40138 Bologna, Italy; maria.iorio@studio.unibo.it (M.I.); nikkitha.umesh@gmail.com (N.U.G.); monica.deluise2@unibo.it (M.D.L.); giuseppe.gasparre3@unibo.it (G.G.); 2Center for Applied Biomedical Research, University of Bologna, 40138 Bologna, Italy; annamaria.porcelli@unibo.it; 3Centro Studi e Ricerca sulle Neoplasie Ginecologiche (CSR), University of Bologna, 40138 Bologna, Italy; 4Department of Pharmacy and Biotechnology (FABIT), University of Bologna, 40126 Bologna, Italy; 5Interdepartmental Center of Industrial Research (CIRI) Life Science and Health Technologies, University of Bologna, 40064 Ozzano dell’Emilia, Italy

**Keywords:** cancer metabolism, tumor microenvironment, metabolic reprogramming, cancer-associated fibroblasts, tumor-associated macrophages

## Abstract

**Simple Summary:**

Along with cancer cells, tumor mass also contains numerous types of non-malignant cell populations, all together contributing to an organ-like multicellular organization. This review underlines the importance of taking into consideration the effects metabolic drugs used for cancer therapy may exert on the cells of the tumor microenvironment.

**Abstract:**

Metabolic reprogramming is a well-known hallmark of cancer, whereby the development of drugs that target cancer cell metabolism is gaining momentum. However, when establishing preclinical studies and clinical trials, it is often neglected that a tumor mass is a complex system in which cancer cells coexist and interact with several types of microenvironment populations, including endothelial cells, fibroblasts and immune cells. We are just starting to understand how such populations are affected by the metabolic changes occurring in a transformed cell and little is known about the impact of metabolism-targeting drugs on the non-malignant tumor components. Here we provide a general overview of the links between cancer cell metabolism and tumor microenvironment (TME), particularly focusing on the emerging literature reporting TME-specific effects of metabolic therapies.

## 1. Basics of Metabolic Reprogramming in Cancer

The concept that tumors have specific metabolic requirements has been known since the seminal Warburg’s observation of the aerobic glycolysis in cancer tissue [1]. High glucose uptake and upregulation of glycolysis in cancer sustains the biosynthesis of macromolecules which serve as building blocks of daughter cells [2]. As a consequence, lactate is secreted to the surrounding environment, to avoid lethal intracellular levels of acidification. Besides glucose, cancer cells also use glutamine via glutaminolysis to support production of macromolecules, since this metabolite serves as the main carbon and nitrogen donor for nucleotide synthesis and anaplerotic reactions of the tricarboxylic acid cycle (TCA) [3]. In parallel, most cancers still preserve mitochondrial respiration to produce energy, ensure redox homeostasis and guarantee appropriate extracellular oxygen sensing [4,5,6]. However, oxidative phosphorylation (OXPHOS) rate and activity of individual electron transport chain (ETC) complexes may vary in cancer cells, contributing to generation of elevated reactive oxygen species (ROS) concentrations, often associated with high tumor aggressiveness and poor prognosis [7,8]. Apart from reprogramming of glycolytic and OXPHOS rates, rapidly proliferating cancer cells also show modification of lipid metabolism. For example, activation of de novo synthesis of fatty acids is often observed in cancer, among others, to provide lipids for membrane formation. Thus, many cancers exhibit overexpressed fatty acid synthase (FASN), a key biosynthetic enzyme involved in lipogenesis [9,10,11].

The signaling pathways regulating cellular metabolism are mainly orchestrated by mammalian Target Of Rapamycin Complex I (mTORC1), under control of MAPK or PI3K/AKT cascades, which are often activated in cancer as a consequence of oncogene and tumor suppressor mutations. mTORC1 signaling increases glycolysis via upregulation of Hypoxia-Inducible Factor 1 alpha (HIF-1α) and c-Myc; supports glutamine metabolism and anaplerosis by repressing sirtuin 4, and promotes lipid biosynthesis and the pentose phosphate pathway by activating Sterol Regulatory Element-Binding Protein 1 (SREBP-1) [12,13]. On the other hand, mTORC1 antagonist AMP activated kinase (AMPK) guards cellular energy homeostasis by slowing down biosynthetic reactions upon increased AMP/ATP ratio in low nutrient conditions [14].

It is important to note that, in some cases, metabolic reprograming of cancer cells is a direct consequence of the genetic hit driving the transformation, as in the case of acute myeloid leukemia, glioma, cholangiocarcinoma and chondrosarcoma, where mutations in TCA cycle enzyme isocitrate dehydrogenases (IDH) are responsible for cancer progression [15]. In particular, mutated forms of either cytosolic (IDH1) or mitochondrial (IDH2) isoforms acquire neomorphic function by which α-ketoglutarate (α-KG) is reduced to D-2-hydroxyglutarate (D-2HG), an oncometabolite whose accumulation eventually triggers a series of epigenetic modifications promoting tumor growth [16].

These are only some of the most emblematic examples of cancer metabolic reprogramming and the complexity of cancer metabolism has extensively been reviewed elsewhere [17,18,19].

## 2. Non-Malignant Cell Populations in Tumor Microenvironment

Tumor microenvironment (TME) is characterized by a specific acellular milieu, with variable pH, extracellular matrix protein composition, oxygen and nutrient concentrations, but the term is also used when referring to the non-malignant cells found in tumor mass, which along with cancer cells form an organ-like multicellular organization. These include (i) endothelial cells that form the walls of tumor vasculature, allowing nutrient, but also chemotherapy delivery; (ii) stromal cells, mainly represented by cancer-associated fibroblasts (CAFs), which support cancer progression by supplying, among others, extracellular matrix proteins [20]; (iii) several cell types of innate and acquired immunity, whose interchange between inflammatory and wound-healing functions respectively decreases or promotes neoplastic progression; and (iv) cells of the cancer associated parenchyma (CAPs), recently recognized as another functionally relevant component of progressing tumor mass [21]. The effect of different TME cell types on cancer cell progression may vary. For example, pancreatic cancer is particularly dependent on certain types of CAFs [22], whereas neutrophils have been found to exert a crucial role in the homing of breast cancer metastases [23]. Even though, in principle, the immune system should recognize mutation-derived, novel cancer cell surface epitopes as foreign, tumors are known to avoid the cytotoxic inflammatory effects by attracting immune cells similar to those involved in wound healing [24]. As a result, immune cells in TME often inopportunely act as a support to tumor growth, rather than as a defense mechanism. In a simplified view, the cytotoxicity of CD4+, CD8+ T effectors and natural killer (NK) cells is subdued in tumor tissue to reduce cancer cell clearance, whereas the immunosuppressive myeloid-derived suppressor cells (MDSCs) and Foxp3-expressing regulatory T-cells (Tregs) are accumulated [25]. Similarly, macrophage polarization from inflammatory (anti-tumorigenic) M1 to proangiogenic M2 tumor-associated macrophages (TAMs) is most often observed in cancer [26,27]. Pro- or anti-tumorigenic effects have also been described for various populations of neutrophils, dendritic cells and B lymphocytes. Many of the mechanisms linking tumor progression and TME components have been identified and are even being successfully exploited for anti-cancer therapy. The most well-known phenomenon regards cancer cell overexpression of programmed death ligand 1 (PD-L1), which causes cytotoxic T-cell exhaustion and immunosuppression, allowing cancer cells to escape immune system surveillance [28]. Moreover, transformation to malignancy is known to change the cancer cell secretome, which often includes a series of immunosuppressive (TGF-β, IL-10), proangiogenic (VEGFA) and CAF inducing (PDGFs, FGF2) growth factors and cytokines. However, less is known on how cancer cell metabolic status and its resulting by-product metabolites affect TME, and how this modifies the efficacy of metabolism-targeting therapies.

## 3. The Crossroads between Cancer Metabolic Reprogramming and TME

Recent years have seen an outburst of literature recognizing the role of TME cell populations in defining cancer cell metabolism. In particular, cancer cells’ avidity for glucose and amino acids results in reduced availability of these metabolites in the tumor mass. Such impoverished environment has shown to profoundly modulate the phenotype of non-malignant cells in the TME, mainly skewing their functions to sustain tumor growth [24,29,30]. As a consequence of metabolite shortage in the tumor extracellular milieu, a series of biochemical relationships are developed between cancer cells and non-malignant cell populations, which apart from competition for available nutrients, may include symbiosis and parasitosis-like phenomena (Figure 1).

### 3.1. Cancer Cell-TME Competition

Metabolic competition is particularly relevant for the fate of immune cells in TME, as their inflammatory, anti-tumor activity is fueled by the same metabolites used by fast-proliferating cancer cells (Figure 1). For example, glucose uptake is essential for T-cell proliferation [31], and supports the maintenance of T-cell inflammatory functions [32,33]. Similarly, cytotoxic activity of NK cells depends on glucose metabolism [34], as well as functioning of M1 macrophages [35]. Thus, in general, glucose deprivation suppresses anti-tumor immunity. On the same note, amino acid depletion in the tumor extracellular milieu is also associated with immunosuppression [36]. For example, serine seems to be essential for cytotoxic T-cell proliferation [37], whereas glutamine deprivation was found to induce Treg phenotype [38]. These observations suggest that patients with tumors particularly avid for glucose or amino acids would probably not be the optimal candidates for immunotherapy, implying that, in general, metabolic status of a cancer cell affects patient response. Indeed, one study reports that advanced melanoma patients who respond to immunotherapy have tumors with high mitochondrial activity, as indicated by proteome enrichment of pathways such as TCA, fatty acid oxidation and ketone body metabolism [39]. It may be hypothesized that the active oxidative metabolism in such melanoma cells, presumably less dependent on glucose than glycolytic melanomas, might reduce nutrient competition with the T lymphocytes, eventually leading to higher antigen presentation and IFN signaling in the responding patients, making these tumors vulnerable to T-cell cytotoxic activity [39]. This implies evaluation of tumor metabolic profile should be considered as an additional factor for predicting the response to immunotherapy.

### 3.2. Metabolic Symbiosis between Cancer and TME

Whereas cells of the immune system are mainly competing for nutrients, CAFs have been described to metabolically co-operate with cancer cells. For example, in breast and prostate cancer models, CAFs were found to produce lactate which is then used by cancer cells, resulting in a symbiotic relationship (Figure 1) [40,41]. Similarly, in an ovarian cancer model, stromal cells were found to supply glutamine in nutrient-deprived conditions (Figure 1) [42]. Interestingly, metabolic liaison between neoplastic and neighboring non-malignant cells is, in general, proving to be particularly important during metastatic homing, since the host organ metabolic status may provide beneficial properties to a metastatic cancer. For instance, ovarian cancer metastases are preferentially found in adipocyte-rich momentum, which then, by lipolysis, fuels metastatic outgrowth supporting energy production via beta-oxidation in cancer cells [43]. Preventing such supportive metabolic environment in metastatic tissues is an intriguing target for developing anti-cancer therapies.

### 3.3. Cancer Cells as Parasites Feeding from the Surrounding Non-Malignant Neighbors

Interestingly, impoverished environment may also cause establishment of parasite-like relationship, such as when cancer cells induce autophagy in neighboring CAFs to obtain stroma-derived nutrients [44]. In particular, pancreatic ductal adenocarcinoma cells were described to stimulate autophagy in pancreatic stellate stromal component to produce alanine required for cancer cell growth in low nutrient conditions (Figure 1) [45]. Moreover, the parasitic activity of cancer cells may affect not only their neighboring tissue, but was shown to condition the distant metastatic niche from the primary tumor site [46]. Specifically, breast cancer-secreted miR-122 was shown to downregulate glycolytic rate of lung fibroblasts and astrocytes, increasing the local pool of glucose available for the incoming metastatic cells (Figure 1) [46].

### 3.4. By-Products of Cancer Cell Metabolism as Modifiers of TME Phenotype

Apart from causing nutrient depletion, cancer cell metabolic reprogramming also results in an atypical accumulation of certain by-product metabolites. These may be used in symbiotic relationship with the stroma, but have also been described to modulate non-malignant cell phenotype, often leading to immunosuppressive effects (Figure 2). The most studied cancer metabolic by-product is lactate [47]. In recent years, it has been acknowledged that, rather than just being a metabolic waste, lactate is actually a signaling molecule, as a recent study showed that it leads to M2 polarization by activating macrophage G protein-coupled receptor 132 [48]. In addition, several studies have implicated lactate as a potential respiratory fuel in tumors. By administration of 13C-lactate into human non-small-cell lung cancers in patients, lactate was found as a predominant carbon source for the TCA cycle [49]. Moreover, lactate was shown to support the survival of breast cancer cell lines under glucose depletion [50]. Generally, a bulk of data is available that convincingly attributes several TME-related pro-tumorigenic roles to lactate. It was shown to induce angiogenesis, either by directly activating endothelial cells [51], or indirectly by fostering secretion of proangiogenic growth factors by cancer cells [52] and TAMs [53]. Moreover, lactate secretion leads to suppression of anti-tumor immune response, by inhibiting NK and T-cell cytotoxic functions [54], increasing MDSC numbers [55], favoring Treg differentiation [56], increasing PD-L1 expression in cancer cells [57], and promoting M2 polarization in TAMs [58,59].

There are several other cancer cell metabolic by-products which modulate TME. For example, glioblastoma cells were recently shown to secrete branched-chain ketoacids (BCKAs), whose uptake by in vitro stimulated macrophages reduced phagocytosis, allowing the authors to hypothesize that the BCKAs may also contribute to immunosuppression [60]. On a similar note, increase in tryptophan degradation by indoleamine-2,3-dioxygenase (IDO) results in abnormal accumulation of kynurenine, which binds to the Aryl Hydrocarbon Receptor in T-cells and promotes expression of genes involved in differentiation to Treg phenotype, thus promoting tumor growth [61]. Elevated kynurenine levels have also been shown to inhibit cytotoxic T- and NK cells’ expansion [62]. Similarly, polyamines derived from elevated arginine conversion to ornithine correlate with downregulation of inflammatory cytokines, suggesting their immunosuppressive effect [63], and the breakdown of adenine in low oxygen conditions causes adenosine accumulation in hypoxic areas of certain tumors, where it inhibits cytotoxic capacity of the immune system [64].

Furthermore, in parallel to the pro-tumorigenic epigenetic modifications in cancer cells [16], D-2HG was found to be secreted by glioblastoma cells and then uptaken by the T lymphocytes, leading to downregulation of their cytotoxic signature and causing immunosuppression associated to polyamine biosynthesis [65,66]. Similarly, glioblastoma setting was associated also with the TME-related pro-tumorigenic effect of glutamate, which was shown to promote Treg proliferation, activation and suppressive function [67]. However, in a different context, glutamate was reported to stimulate inflammatory cytokine production and proliferation of activated T-cells [68], suggesting not all cancer metabolism by-products necessarily lead to immunosuppression. Indeed, it is important to acknowledge that the same metabolite may have opposing effects on different TME cells. For example, glutamine has been reported as a fuel for polarization towards protumorigenic M2 phenotype of macrophages [69,70], meaning that deprivation of glutamine in the tumor extracellular milieu may, on one hand, suppress cytotoxic T-cell functions [38], but prevent pro-tumorigenic activity of M2 macrophages. Most likely, the final effect of a metabolite on TME is highly influenced by the resident tissue metabolic status, as the glutamate-rich brain tissue [71] undoubtedly sets different selective pressures on cancer cells than the lipid-rich ovarian milieu [43].

## 4. Current Metabolism-Based Therapeutic Strategies

Noteworthy, one of the oldest chemotherapies for cancer is based on targeting cancer metabolism. Antifolates, such as methotrexate (MTX), used since the 1940s, inhibit one-carbon donors essential in the synthesis of purines, pyrimidines, serine and methionine— all critical metabolites for DNA synthesis. By inhibiting the key enzymes in folate metabolism, drugs such as MTX induce folic acid deficiency, hindering DNA synthesis and cell division [72]. Today, numerous metabolism-based anti-cancer approaches exist, either already applied in therapeutic regimes, or being tested in clinical and preclinical studies (Figure 3) [73,74].

### 4.1. Targeting mTORC1 Signaling

Since the gene expression profile involved in cancer cell metabolic reprogramming is mainly orchestrated by the oncogenic activation of mTORC1, rapamycin analogues temsirolimus and everolimus have been used for treatment of neoplasias such as renal cancer [75]. However, mTORC1 inhibitors have generally shown a somewhat disappointing efficacy in clinical trials, and efforts are being made to develop a more efficient, new generation of mTORC1-targeting drugs [76,77,78].

### 4.2. Preventing Cancer Cell Lactate Secretion as a Promising Therapeutic Strategy

Rather than targeting a general pathway as mTORC1, inhibiting specific metabolic reactions has been showing promising results. For example, since lactate’s role as a mere “metabolic waste” has been upgraded, acknowledging its signaling activity and potential as an energy source, targeting lactate dehydrogenase (LDHA) and monocarboxylate transporters (MCT), which mediate lactate secretion/consumption, has been gaining interest in cancer therapy [49,79]. In particular, disrupting MCT1 activity by AZD3965 increased apoptotic cell death and decreased viability of Non-Hodgkin’s lymphomas [80], and a first in human clinical trials was started in advanced solid tumors to evaluate the efficiency and toxicity of the drug and is still ongoing ([81] NCT01791595).

### 4.3. Inhibitors of the TCA Cycle Enzymes

Moreover, targeting TCA cycle enzymes has been showing promising results in clinics. Indeed, the currently most successful therapeutic approach based on metabolic reprogramming regards the inhibitors of the mutant IDH1/2 [82,83,84,85], which are being used to treat acute myeloid leukemia patients, as well as gliomas and other solid cancers, such as cholangiocarcinoma and chondrosarcoma [15]. Another intriguing approach targeting TCA cycle regards the simultaneous inhibition of both pyruvate dehydrogenase (PDH) and alpha-ketoglutarate dehydrogenase complex (KGDH) with the same anti-cancer compound, namely lipoate derivative, CPI-613 (devimistat). By blocking both glycolysis and TCA cycle, this drug was found to induce cell death in pancreatic cancer patients and non-small-cell lung tumor xenograft models, with minimal side effects to normal cells [86,87]. The latter clinical trial is still active, with encouraging complete as well as partial response rates reported up to date ([86] NCT01835041).

### 4.4. The Anti-Cancer Effects of OXPHOS Inhibitors

Apart from targeting glycolytic and TCA cycle enzymes, the recent acknowledgment of OXPHOS maintenance in cancer has led to exploration of anti-cancer strategies against mitochondrial respiration. In this context, respiratory complex I (CI) has been recognized as a promising metabolic anti-cancer target, due to its role in maintaining fast proliferating cancer cell redox homeostasis, aspartate metabolism and hypoxic-adaptation [88,89]. In particular, the FDA-approved and well-tolerated CI inhibitor metformin, traditionally used to lower glycemia in diabetic patients, has displayed anti-tumorigenic effects in both experimental settings and in clinical studies, suggesting beneficial aspects of metformin repurposing for cancer treatment [90,91]. Moreover, additional compounds are being developed that corroborate the anti-tumorigenic properties of CI inhibition and are being tested in clinical trials (BAY 87-2243 [92], IACS-010759 [93], IM156 (NCT03272256)). Inhibition of other OXPHOS complexes has also been taken into consideration. For example, atovaquone, a complex III specific inhibitor has been shown to decrease the propagation of MCF7-derived cancer stem cells (CSCs), in which, apart from reducing OXPHOS activity, mitochondrial mass, membrane potential and affected ROS production [94]. Of note, despite being commonly called for in various contexts of cancer biology, the possibility of targeting ROS by antioxidants as an anti-cancer therapy remains controversial. On one hand, treatment with compounds such as glutathione (GSH) precursor N-acetylcysteine (NAC) reduce ROS genotoxic activity. On the other hand, in an already transformed cell, which risks ROS-mediated toxicity, the antioxidant effect may result in a protective for cancer [95].

### 4.5. Targeting Amino Acid Metabolism

Many of the metabolism-based anti-cancer therapies target amino acid metabolism modulators. Since glutaminases (GLSs) are key enzymes allowing extracellular glutamine to be used for anaplerosis in cancer cells, with mitochondrial GLS particularly overexpressed in several types of cancer [3], GLS inhibitors, such as CB-839, are currently being tested in clinical trials (NCT02071862, NCT02071888, NCT02071927 and [96] NCT02861300) [97]. In particular, the study NCT02861300 reports interesting results indicating that CB-839 may be efficient in improving response to 5-FU in PIK3CA-mutant colorectal patients [96]. Furthermore, knockdown of asparagine synthetase, the enzyme that synthesizes asparagine de novo from aspartate and glutamine, leads to cell death, suggesting an essential role of asparagine in maintaining cancer cell viability [98]. Thus, L-asparagine-depleting enzyme L-ASN is currently being used for leukemia treatment [99], and its efficacy is being investigated in the context of certain solid tumors [100]. Similarly, arginine-depleting enzymes are being tested in clinical trials for advanced solid cancer treatment (NCT03254732, [101] NCT02029690), since arginine was identified as another critical cancer cell nutrient [102,103], and is also known to promote M2 macrophages phenotype [104]. In particular, the study NCT02029690 showed promising data in terms of tumor size regression in arginine inhibitor group [101].

### 4.6. Blocking Lipid Metabolism in Cancer

Finally, strategies to inhibit lipid synthesis in cancer have also shown promising results. In particular, FASN inhibitors, such as cerulenin, C75, orlistat, TVB-2640, TVB-3166, C93 and naturally occurring polyphenols are being investigated for their anti-tumorigenic efficacy [45,105,106,107,108]. For example, cerulenin was found to inhibit FASN in breast cancer cell lines and induce delay in disease progression of a xenograft ovarian cancer model via cytotoxic activity [109,110].

### 4.7. Accounting for Cancer Cell Metabolic Plasticity

Even though most of the listed compounds here show considerable promise in clinical trials, it must be acknowledged that metabolic drugs are rarely used as a stand-alone protocol and are instead currently considered a valid adjuvant approach to the standard anaplastic therapy. This is mainly due to cancer cell metabolism being a dynamic property, shaped by external pressures, which dramatically change between early and late events of primary tumor growth, and even more so during metastatic processes. Its high adaptability must be taken into consideration in order to efficiently exploit metabolic reprogramming for anti-cancer therapy. Most likely, using compound cocktails targeting complementary biochemical reactions or setting up simultaneous calorie restriction regimes will lead to satisfactory results [111,112,113], provided that the consequences of metabolic drugs on TME are also taken into account.

## 5. Targeting Cancer Metabolism Alters the Non-Malignant Cells of the TME

The effects of cancer-related metabolic drugs on non-transformed TME components depend on both direct and indirect mechanisms, as well as on systemic response to the drug, the latter being especially critical in the context of recruitment of non-resident populations, such as immune cells.

It is interesting to note that certain drugs targeting metabolic reprogramming of cancer cells are actually designed to ensure a proper functioning of non-malignant populations of TME. For example, IDO inhibitors currently under clinical development (NLG919, NLG2101) are used to block tryptophan degradation. Their purpose is to maintain sufficient microenvironment concentrations of this essential amino acid and prevent kynurenine accumulation, ensuring the efficient anti-tumor immune response by cytotoxic T lymphocytes, which are particularly sensitive to tryptophan/kynurenine levels [114,115]. Another metabolism targeting approach designed to prevent immunosuppression regards the reduction of adenosine levels in tumor extracellular milieu, since its high concentrations were found to reduce T-cell cytotoxicity [64,116].

Emerging literature reveals that metabolic drugs not originally intended to affect the cells of TME modulate non-malignant cell functions, which in turn has consequences on tumor progression (Table 1).

Often, metabolic drugs display synergic anti-tumorigenic effects by acting on both cancer and TME. For example, GLS inhibition by either Compound 968 or BPTES decreased arginase (Arg1) expression in mouse bone marrow-derived macrophages (BMDMs) and triggered M1 signature [69]. Even though this phenomenon warrants investigation in the context of cancer, it may be hypothesized that drugs like Compound 968 may skew TAM polarization towards the M1 phenotype, adding up to the anti-tumorigenic effect of GLS targeting. Similarly, 5-fluorouracil and gemcitabine have been shown to enhance the anti-tumor immune activity, by selectively depleting MDSCs in vivo [128,129]. In particular, these drugs target the splenic MDSCs, which then facilitates the immune invasion, to enhance IFN-gamma production by tumor-specific CD8(+) T-cells infiltrating the tumor, promoting the T-cell-dependent anti-tumor responses. Moreover, gemcitabine was also shown to improve NK cell activity [130]. On the other hand, in some cases, metabolic therapies designed to prevent cancer cell progression, may have undesirable TME effects, which allow the tumor to adapt to the selective pressures triggered by the treatment. We have recently identified an association between targeting CI and macrophage-mediated survival response [131]. Moreover, CI depletion in cancer seems to trigger various TME-related consequences, which we have recently reviewed elsewhere [132].

It is important to consider that the same drug may have diverse effects on different TME populations, and even opposite effects on the same cell type in different contexts, as it was reported for metformin [132]. On the same note, mTORC1 inhibition was found to elicit both anti- and pro-tumorigenic effects on TME. In particular, the treatment of colorectal cancer cells with mTORC1 inhibitor rapamycin was associated with vessels displaying blunt ends, few connections and abrupt changes in diameter, a phenomenon which correlated with impaired VEGF production. Rapamycin markedly reduced VEGF-induced HUVEC proliferation in vitro, in a dose-dependent manner [126], suggesting targeting mTORC1 constrains tumor progression not only by acting on cancer cells, but also through an anti-angiogenic activity. Conversely, mTORC1 inhibition reduced cytotoxic T-cell activation [133] and promoted Treg phenotype [134], suggesting mTORC1 inhibition may promote immunosuppression and accelerate tumor growth. In line with this hypothesis, mTORC1 activation seems to be essential also for NK activity [34], implying that mTORC1 inhibitors would compromise their cytotoxic function. Thus, limited efficacy of mTORC1 inhibitors reported in clinical trials could potentially be explained, among others, by rapamycin-mediated suppression of anti-tumor immunity.

Another example of how targeting cancer metabolism may lead to unintended immunosuppressive effects arrives from targeting DNA synthesis by MTX, which has been associated with adenosine accumulation in glioblastoma TME, and with a subsequent immunosuppressive response [123]. Thus, it seems that the MTX efficacy could be potentiated by parallel inhibition of adenosine-mediated immunosuppressive effects. However, the MTX-mediated TME alterations are still not fully understood, as it was reported to cause both reduction in T effector cell number [135] and suppression of the Treg functions [136].

While the majority of the current literature focuses on analysis of the immune TME portion, it is important to underline that there are several other cell populations present in the tumor mass, whose contribution is being often neglected. An example of metabolic inhibitors affecting non-immune cells of TME regards GLS inhibitors, which have been associated with synergistic anti-tumorigenic effects, since apart from suppressing cancer cell survival [137], reduced proangiogenic signals deriving from connective tissue. In particular, GLS inhibitor Compound 968 appears to reduce tumor-derived extracellular vesicles (TEVs), which in turn decreases myofibroblastic differentiation of adipose-derived stem cells (ASCs), leading to decreased pro-angiogenic capacity [118]. Specifically, the reduced TEV outbudding/release from human metastatic breast cancer cells prevented TGF-beta-mediated activation of ERK1/2 and JNK1/2 required for myofibroblastic differentiation in ASCs [138]. Similarly, the treatment with antioxidant NAC was shown to inhibit CAF pro-tumorigenic role in breast cancer models, by preventing CAF-mediated lactate secretion and in turn decreasing nutrient supply for cancer cells [124,139].

Apart from the connective tissue, metabolic drugs have been shown to also affect endothelial cell functions. For example, FASN inhibitor Orlistat was associated with reduced length and area of peritumoral blood vessels, and with preferential secretion of anti-angiogenic isoform VEGFA165b in the melanoma B16-F10 lung colonization model [125]. Moreover, orlistat and cerulenin were shown to differentially modulate VEGF-C and -D, regulators of lymphatic vessel density [117]. These results suggest that FASN inhibitors, in parallel to reducing cancer cell proliferation by targeting lipid metabolism, may also modulate lymphatic endothelial cells’ status. On the same note, L-ASN was demonstrated to exhibit anti-angiogenic activity through its direct effect on vascular remodeling [121]. L-ASN treatment caused a marked reduction in the number and complexity of HUVEC tube-like structures. This was accompanied by total protein reduction and thinning of HUVEC stress fibers, indicating that L-ASN impedes morphogenesis of endothelial cells into capillary-like structures [121]. A similar phenomenon was reported by another group, who showed that L-ASN inhibited endothelial cell differentiation into tubes on Matrigel in a dose-dependent fashion [122].

Even though the data regarding the unintended effects of metabolic therapies on TME cells are still scarce, it is reasonable to hypothesize that more than one TME population may be affected, simultaneously contributing to the final outcome on tumor progression. This is well exemplified by the case of arginine depletion therapy. Since certain cancer cells may become auxotrophic for arginine, its enzymatic depletion from TME is being evaluated as an anti-cancer approach [103]. However, arginine is essential also for T-cell survival [140], and it is extensively catabolized by M2 macrophage and MDSCs as well [141], meaning that such approach may unintentionally lead to T-cell immunosuppression on one hand, and to inhibition of pro-tumorigenic monocytic TME components on the other. Thus, to optimize the anti-tumorigenic effects, the choices for metabolic therapies should be guided by a precise pre-evaluation of tumor immunophenotype and general TME composition.

## 6. Conclusions

Targeting cancer metabolism has been considered as a somewhat generalizable approach to cancer treatment, as highly proliferating solid cancers often share common targetable routes of metabolic reprogramming. However, the additional level of complexity given by the TME populations and by their alteration in response to metabolic therapy, calls attention to the fact that even the most general approach may require a personalized fine-tuning in terms of drug delivery and dosage, and possibly implementation of adjuvant therapies targeting TME. In this context, it is important to underline recent advances in nanotechnology [142], leading to development of approaches aimed for drug delivery to specific TME populations. For example, nanoparticles carrying adhesion proteins of T- cells [143] or cytokine encoding-plasmids [144] were engineered to confer immunosuppression escape by site-specific release of anti-cancer drugs. Even targeting extracellular stroma-derived matrix, by loading nanoparticles with collagenase, resulted feasible and successful in enhancing drug diffusion and efficacy in pancreatic tumors [145].

Thus, since nanoscale formulations targeting specific cell tumor populations are expected to be available in the close future [146] a comprehensive pre-evaluation of the immunophenotype and stromal proportion of a tumor should be considered in order to guide also metabolism-based therapeutic choices. Preclinical cancer drug testing should include not only the evaluation of the efficient anti-tumorigenic effect on the cancer cell and estimation of the toxic effects on the organism, but also consider the consequences that the treatment might have on endothelial cells, resident/recruited immune cells, and the cancer-associated mesenchymal component, a concept applicable to any cancer therapy. In this context, while the encouraging effects of immunotherapy has prompted the majority of the current oncologic clinical trials to evaluate T-cell count and phenotype, other TME cell populations are rarely taken into consideration.

We predict that the evidences described in this review are only the tip of an iceberg, and that in the following years more attention will be drawn towards the comprehension of the effects metabolic therapies have on TME. Novel in vivo metabolism-imaging approaches, as well as elaborate organotypic 3D co-culture models are now available to deliver these new discoveries.

## Figures and Tables

**Figure 1 cancers-13-05447-f001:**
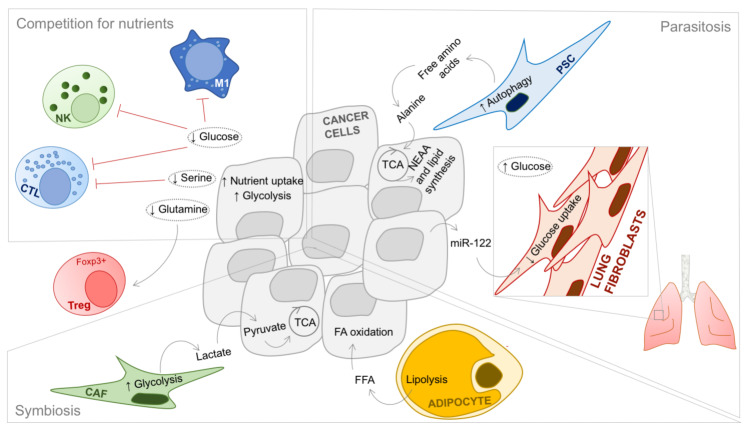
Metabolic relationships between cancer and non-malignant cells of the tumor microenvironment. Competition for nutrients occurs mainly between cancer and cytotoxic immune cells. In particular, cancer cells (gray) uptake high amounts of glucose to sustain elevated glycolytic rate. At the same time, glucose is required for proliferation and maintenance of inflammatory functions of cytotoxic T lymphocytes (CTL), cytotoxic activity of natural killer cells (NK), and for functioning of proinflammatory macrophages (M1). Similarly, cancer cells uptake high amounts of amino acids, which have been found as essential, among others, for CTL cytotoxic activity, as in the case of serine. Thus, competition for nutrients generally leads to immunosuppression (red lines) and reduced tumor clearance, also because the resulting impoverished extracellular milieu may induce pro-tumorigenic immune cell phenotype, as in the case of glutamine deprivation associated with regulatory T cell (Treg) development. Metabolic symbiosis most often occurs between cancer cells and the residents of the affected tissue. For example, breast cancer may uptake lactate released as waste from highly glycolytic activated cancer-associated fibroblasts (CAF), to feed tricarboxylic acid cycle (TCA) and support oxidative metabolism. On the other hand, ovarian cancer cells were shown to uptake free fatty acids (FFA) released from lipid-rich adipocytes, to generate acetyl-CoA via fatty acid (FA) oxidation. Parasite-like behavior of cancer cells has been, among others, described in pancreatic ductal adenocarcinoma which induces autophagy in pancreatic stellate cells (PSC) leading to free amino acid release. In particular, alanine is then uptaken by cancer cells to support anaplerosis and biosynthetic reactions, such as non-essential amino acid (NEAA) and lipid synthesis. Interestingly, a long-distance metabolic parasitism was reported in the context of breast cancer, where the release of mi-122 conditioned distant organs, such as lungs, to reduce their glycolytic rates and consequently, glucose uptake, leaving higher glucose concentrations available for the metastatic cell to use while nesting in the putative metastatic niche.

**Figure 2 cancers-13-05447-f002:**
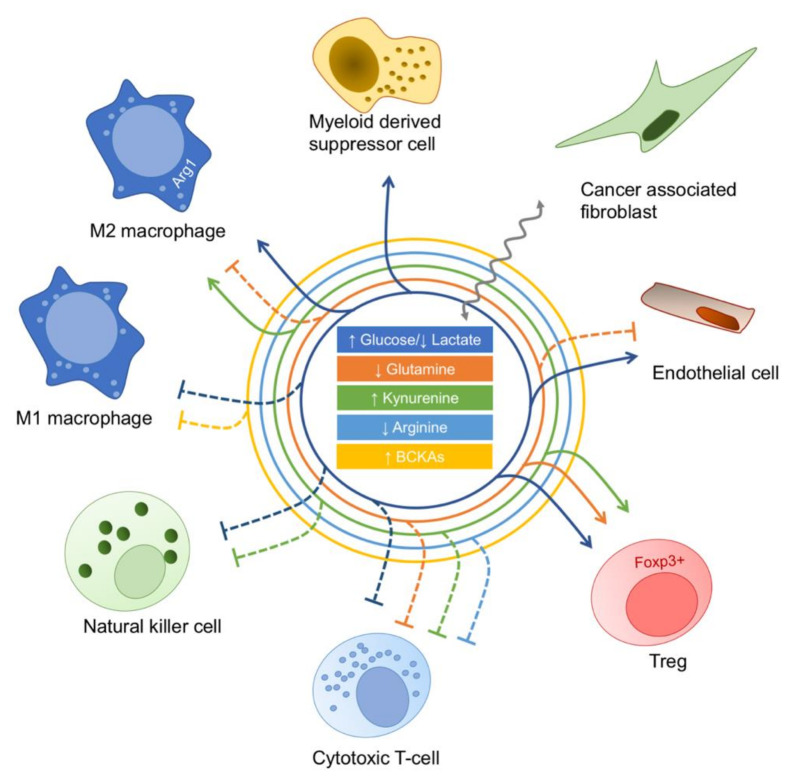
The consequences of cancer cell metabolic reprogramming on the non-malignant populations of TME. Nutrient shortage and increased amounts of cancer cell metabolic by-products modulate the phenotype of TME cell populations. Full and dashed lines indicate, respectively, stimulation and inhibition. Wavy line represents metabolite exchange between cancer-associated fibroblasts and cancer cells.

**Figure 3 cancers-13-05447-f003:**
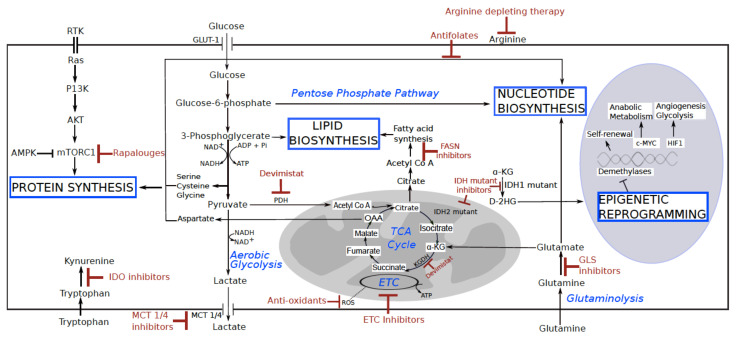
Targeting metabolic reprogramming in cancer. In a fast proliferating cancer cell, energy is most often produced simultaneously by both glycolysis and oxidative phosphorylation. High glucose uptake sustains increased glycolytic rates required for macromolecule biosynthesis (proteins, lipids and nucleotides), furthermore fueled by TCA cycle intermediates that can be replenished by the use of glutamine as carbon and nitrogen source. In certain cancers, the atypical accumulation of oncometabolites, such as D-2HG, may promote epigenetic remodeling. The main cellular pathway regulating metabolic reprograming is mTORC1, whose activity regulates several transcription factors to promote glycolysis and macromolecule biosynthesis. These reactions have been recognized as valid targets for cancer treatment and some of the current metabolic anti-cancer strategies are indicated in red.

**Table 1 cancers-13-05447-t001:** A list of TME-related effects caused by therapies targeting cancer cell metabolic reprogramming.

Drug	Target Enzyme	Cancer Type	TME Cell Type	Effect	Reference
BPTES	GLN	–	BMDMs	Suppressed expression of M2-specific genes and Arg1 activity	[69]
Cerulenin	FASN	Melanoma cells (SK-MEL-25) and tumors (B16-F10)	Endothelial cells (HUVEC)	Anti-angiogenic effect	[117]
Compound 968	GLN	Breast cancer cells (MDA MB-231)	ASCs	Reduced release of TEVs and anti-angiogenic behavior of ASCs	[118]
Compound 968	GLN	–	BMDMs	Suppressed expression of M2-specific genes and Arg1 activity	[69]
IDH-35	IDH	Murine glioma model	CD8+ T-cells	Restoration of cytotoxic T-cell recruitment	[119]
INCB024360	IDO	PDAC tumors (PAN02)	Tregs	Reduced conversion to Treg phenotype	[120]
L-ASN	ASN	Gastric adenocarcinoma cells (AGS)	Endothelial cells (HUVEC)	Anti-angiogenic effect	[121]
L-ASN	ASN	Ovarian cancer cells (HEYA8)	Endothelial cells (HMVEC)	Anti-angiogenic effect	[122]
MTX-LNCs	Folate metabolism	GBM implanted in rats	CD4+ T-cells	Decreased intratumoral cytotoxic T-cells	[123]
NAC	Glutathione	Human breast cancer	CAFs	Suppressed expression of MCT4	[124]
Orlistat	FASN	Melanoma cells (SK-MEL-25) and tumors (B16-F10)	Endothelial cells (HUVEC)	Anti-angiogenic effect; decreased metastases	[125]
Rapamycin	mTORC1	Colon carcinoma cells (CT-26)	Endothelial cells (HUVEC)	Anti-angiogenic effect	[126]
Rapamycin	mTORC1	Human bladder cancer	CD8+ T-cells	Decreased intratumoral cytotoxic T-cells	[127]

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
