# Peer review of "The Neglected Liaison: Targeting Cancer Cell Metabolic Reprogramming Modifies the Composition of Non-Malignant Populations of the Tumor Microenvironment"

_cancers, 2021, doi:10.3390/cancers13215447_

Round 1
Reviewer 1 Report
The Manuscript is a good comprehensive review.
The focus of the review is to target metabolic status of cancer cells in TME and review of literature focused on therapies that are specific for cancer cells in TME environment.
There are not many reviews focused on cancer, metabolic status and TME.
Even though the authors present a comprehensive review, some aspects are missing. Adding them will strengthen the review, please see below.
Section and a picture elucidating the role of Immune cell metabolism is needed, nanoparticles targeting TME subpopulation is missing in the review.
Conclusion is appropriate.
Addition of nanoparticle targeting TME, a good picture elucidating the role of immune cells in TME will strengthen the review.
Reviewer 2 Report
The review by Iorio et al. reports a general overview of cancer cell metabolism and the tumor microenvironment, reporting the current therapies targeting cancer metabolism. It gives an excellent general overview of the topic.
I have only some suggestions about the work. First, it would be interesting to add specific paragraphs about the results of ongoing or concluded clinical trials that the authors cite in the paper.
Also, as a general suggestion, when citing clinical trials, it would be better to add a reference (and not only the number of the trial) if it is available and/or results were published.
On page 8, line 297-298, reference 95 does not refer to cited clinical trials. Ref 95 does not report clinical trials results; I suggest checking or putting the reference before the NCTs numbers.
Line 301-302: Does a more recent work not exist reporting L-asparagine-depleting enzyme L-ASN use for leukemia in humans? Reference 97 is about the use of this molecule on mouse leukemia.
Minor corrections:
The caption of figure 2 reports ‘Figure 1.’
Reviewer 3 Report
The manuscript entitled “The neglected liaison: targeting cancer cell metabolic reprogramming modifies the composition of non-malignant populations of the tumor microenvironment” by Iorio and colleagues highlights the importance of the metabolic reprogramming and the effects of its targeting on the tumor microenvironment. The review presents a detailed overview of the recent literature published on emerging concepts and evidence related to the impact on metabolism-targeting drugs on the composition of non-malignant populations of the tumor-microenvironment.
Overall, this is a clear and well-organized manuscript. I think this paper is excellent and is an important addition to the literature.
